# In situ visualization of endothelial cell-derived extracellular vesicle formation in steady state and malignant conditions

Georgia K. Atkin-Smith[1,2,3,4] ✉, Jascinta P. Santavanond[3,4], Amanda Light[1,2], Joel S. Rimes [1,2], Andre L. Samson [1,2], Jeremy Er[1,2,5], Joy Liu[1,2], Darryl N. Johnson [6], Mélanie Le Page[7], Pradeep Rajasekhar [1,2], Raymond K. H. Yip [1,2], Niall D. Geoghegan [1,2], Kelly L. Rogers [1,2], Catherine Chang[1], Vanessa L. Bryant[1,2,8], Mai Margetts [1], M. Cristina Keightley[3,9], Trevor J. Kilpatrick[10,11], Michele D. Binder[10], Sharon Tran[12,13], Erinna F. Lee[3,12,13], Walter D. Fairlie [3,12,13], Dilara C. Ozkocak[3,4], Andrew H. Wei[1,2,5], Edwin D. Hawkins [1,2,14] ✉ & Ivan K. H. Poon [3,4,14] ✉

Endothelial cells are integral components of all vasculature within complex organisms. As they line the blood vessel wall, endothelial cells are constantly exposed to a variety of molecular factors and shear force that can induce cellular damage and stress. However, how endothelial cells are removed or eliminate unwanted cellular contents, remains unclear. The generation of large extracellular vesicles (EVs) has emerged as a key mechanism for the removal of cellular waste from cells that are dying or stressed. Here, we used intravital microscopy of the bone marrow to directly measure the kinetics of EV formation from endothelial cells in vivo under homoeostatic and malignant conditions. These large EVs are mitochondria-rich, expose the 'eat me' signal phosphatidylserine, and can interact with immune cell populations as a potential clearance mechanism. Elevated levels of circulating EVs correlates with degradation of the bone marrow vasculature caused by acute myeloid leukaemia. Together, our study provides in vivo spatio-temporal characterization of EV formation in the murine vasculature and suggests that circulating, large endothelial cell-derived EVs can provide a snapshot of vascular damage at distal sites.

Endothelial cells are key barrier cells that line the entire vasculature network. Many endothelial cell properties such as cell growth, cytoskeletal reorganization, cell polarity, migration and differentiation depend on highly coordinated cell-to-cell communication[1–7]. One mechanism used by endothelial cells to facilitate intercellular communication is the release of subcellular membrane-bound cargo known as extracellular vesicles (EVs). EVs can be released from live, activated, stressed and dying cells. They include small EVs such as exosomes (typically <200 nm in diameter[8]) and large EVs such as apoptotic bodies, migrasomes and exophers (equal or >1 μm)[8–11]. As EVs can harbour a range of biomolecules including cytokines[12], miRNA[13] and organelles[10], they have been implicated in facilitating intercellular communication[14], antigen presentation[15], viral propagation[16], waste disposal[10,17,18] and inflammatory signalling[12]. Notably, many studies have investigated the formation of small endothelial cell-derived EVs[12,19–24] and demonstrated their role in coordinating cell-to-cell communication[12,25,26]. As endothelial cell health is central to many inflammatory diseases[27], the formation and function of endothelial cell-derived EVs under inflammatory settings is

of significant interest to the field. Notably, inflammatory environments can alter both the quantity and quality of small endothelial cell-derived EVs[12,19,20] and as a result, these EVs exhibit a range of pro-inflammatory properties[19,28].

The formation of large EVs and their clearance by phagocytes in a range of tissues has emerged as an important mechanism for the efficient removal of cellular waste. For example, the generation of apoptotic bodies, a subclass of large EVs released exclusively from apoptotic cells, is essential for the rapid removal of apoptotic materials and limits the release of inflammatory cellular contents[18]. More recently, the generation and clearance of large EVs containing dysfunctional mitochondria (i.e. exophers and migrasomes) from distinct cell types such as cardiomyocytes, neurons and migrating cells were demonstrated as an alternative waste disposal mechanism that maintains tissue homoeostasis, limits inflammation and aids organogenesis[9,10,29–31]. Furthermore, the release of EVs containing healthy mitochondria from spermatid is also a critical process for normal sperm development[32]. Thus, the formation and removal of large EVs carrying organelles is an important mechanism that facilitates systemic cell-to-cell communication and maintains homoeostasis. However, the formation of large, endothelial cell-derived EVs and their subsequent clearance remains to be fully defined.

Although recent studies have visualized the production of EVs in lower-order organisms[29,33–35], performing analogous in situ experiments in murine models remains a major hurdle in the field. To date, few studies have directly captured the generation of EVs in murine models[30,36–38]. Therefore, studies investigating the formation of EVs by endothelial cells are largely restricted to in vitro models that fail to recapitulate the physiological context of the vascular network[28,39,40]. Thus, understanding the in vivo formation, function and clearance of endothelial cell-derived EVs remains a significant gap in the field with broad implications for both steady-state and disease settings. Here, we established an in vivo model to investigate the formation and clearance of large endothelial cell-derived EVs in mice. Using an endothelial cell-specific reporter mouse model (*Vegf-r2*, or Foetal liver kinase 1 (Flk1)-GFP), we detected the presence of large, mitochondria-rich endothelial cell-derived EVs. By performing 4D intravital imaging of the bone marrow (BM) calvarium in live mice, we captured and quantified the generation of endothelial cell-derived EVs in vivo. Analysis by both conventional and imaging flow cytometry approaches demonstrated that various immune cell subsets may interact with and engulf endothelial cell-derived EVs. Furthermore, under inflammatory conditions such as acute myeloid leukaemia (AML), we observed a significant increase in the formation of endothelial cell-derived EVs released from BM vasculature. This observation correlated with the accumulation of these EVs in circulation and endothelial cell loss observed at late-stage disease. Together, these data define the in vivo generation of large endothelial cell-derived EVs under homoeostatic and malignant conditions that may provide a general insight into the health of the vasculature network.

## Results

### Endothelial cells generate large, extracellular vesicles under homoeostatic conditions

To investigate the formation of large endothelial cell-derived EVs in vivo, we established a traceable system using Flk1-GFP mice that express GFP under the control of *vegfr2* that is ubiquitously expressed in endothelial cells[41]. We reasoned that EV-sized particles with high GFP fluorescence are therefore derived from endothelial cells. When examining the blood of Flk1[GFP/+] mice (herein referred to as Flk1-GFP), we could detect a distinct population of FSC low, CD45.2 low, GFP+ particles (indicative of subcellular vesicles that are non-hematopoietic and endothelial cell-derived, respectively) by flow cytometry that were absent in Flk1[+/+] (wt) controls under homoeostatic conditions (Fig. 1a; Supplementary 1). Notably, GFP+ particles were readily detected in

organs such as the BM and spleen in Flk1-GFP mice (Fig. 1b; Supplementary Fig. 1), expressed endothelial cell markers such as CD31, and were low for the platelet marker CD41 (Supplementary Fig. 2a). Furthermore, we could identify EVs that express endothelial cell markers such as CD31, Flk1, CD146 in wt mice in the absence of a traceable fluorescent protein by implementing an antibody-based approach (Supplementary Fig. 2b). We further characterized Flk1-GFP+ particles by performing imaging flow cytometry and confocal microscopy on EVs isolated by fluorescence-activated cell sorting (FACS). These methodologies confirmed Flk1-GFP+ particles as endothelial cell-derived EVs based on their vesicular morphology and subcellular size of ~3 μm in diameter (Fig. 1c, d). As Flk1-GFP expression is generally localized to the cytoplasm, GFP+ EVs are likely to be intact as EV lysis would result in the loss of cytoplasmic GFP signal.

To demonstrate that GFP+ EVs were a result of direct EV production from endothelial cells, we performed intravital microscopy of the BM vasculature in the calvarium as previously described[42–44] (Fig. 1e, f; Supplementary Video 1). Importantly, administration of the vascular dye Evens Blue demonstrates that majority of all blood vessels within the BM calvarium are lined with Flk1-GFP+ endothelial cells (Supplementary Fig. 3a). We collected time-lapse microscopy data of the BM vasculature in Flk1-GFP mice at a temporal resolution of 2–3 min for 2–4 h for each imaging session, and captured the formation of endothelial cell-derived EVs in situ. These data demonstrated that Flk1-GFP+ EVs were directly released from endothelial cells lining the blood vessels into the vasculature (Fig. 1g; Supplementary Fig. 4; Supplementary Videos 2–6). Consistent with ex vivo analysis of EV distribution in organs (Fig. 1b), Flk1-GFP+ EVs could be captured both as they circulated throughout the vasculature, as well as in the BM microenvironment (Fig. 1h). As such, EVs could also be observed on occasion interacting with the blood vessel endothelium and distinct endothelial cells (Supplementary Video 7). The Flk1-GFP+ EVs observed by intravital microscopy were ~3 μm in diameter (Fig. 1i), similar to the size range as determined by ex vivo analysis (Fig. 1d). Importantly, we consistently captured the generation of EV formation in independent imaging experiments. Quantification of EV production demonstrated that this biological process occurs at a rate of approximately once per 2 h of imaging within an 838 μm × 1250 μm tiled imaging area in the calvarium (Fig. 1j). Moreover, we extrapolated the total (predicted) number of EV formation events occurring in the total mouse bone marrow, which predicts that 88 EVs are generated every hour, accounting for over 2000 EVs every day (Supplementary Fig. 3b, c).

EVs can harbour a broad range of biomolecules that can aid intercellular communication as well as intracellular contents that can elicit an inflammatory response if released extracellularly[10,45]. Thus, we examined if the described endothelial cell-derived EVs could contain inflammatory damage-associated molecular patterns (DAMPs) such as DNA or mitochondrial content by flow cytometry. Approximately 30% of Flk1-GFP+ EVs contained nuclear DNA (Hoechst 33342+) and ~80% contained mitochondrial content based on Tom20 (a mitochondria-specific protein) and MitoTracker Red staining (Fig. 1k–m). Notably, Flk1-GFP+ EVs contained polarised mitochondria as treatment of Flk1-GFP BM with the mitochondria uncoupling agent CCCP significantly reduced MitoTracker Red staining (Fig. 1n). Furthermore, we visualized the presence of mitochondria within Flk1-GFP+ EVs by performing Airyscan confocal microscopy on isolated EVs ex vivo (Fig. 1o, Supplementary Video 8). EV analysis by both conventional and imaging flow cytometry approaches demonstrated that Flk1-GFP+ EVs exposed the characteristic cell death marker and 'eat-me' signal, phosphatidylserine (PtdSer) (Fig. 1p, q). These data suggest that Flk1-GFP+ EVs may represent either endothelial cell-derived apoptotic bodies (typically >1 μm, nuclear DNA+/−, PtdSer+, formed in an apoptosis-dependent manner[11]) and/or exophers (~3 μm, mitochondria rich, PtdSer+, formed in an autophagy-dependent manner[10,31]). To examine whether Flk1-GFP+ EVs could be derived from either apoptotic and/or

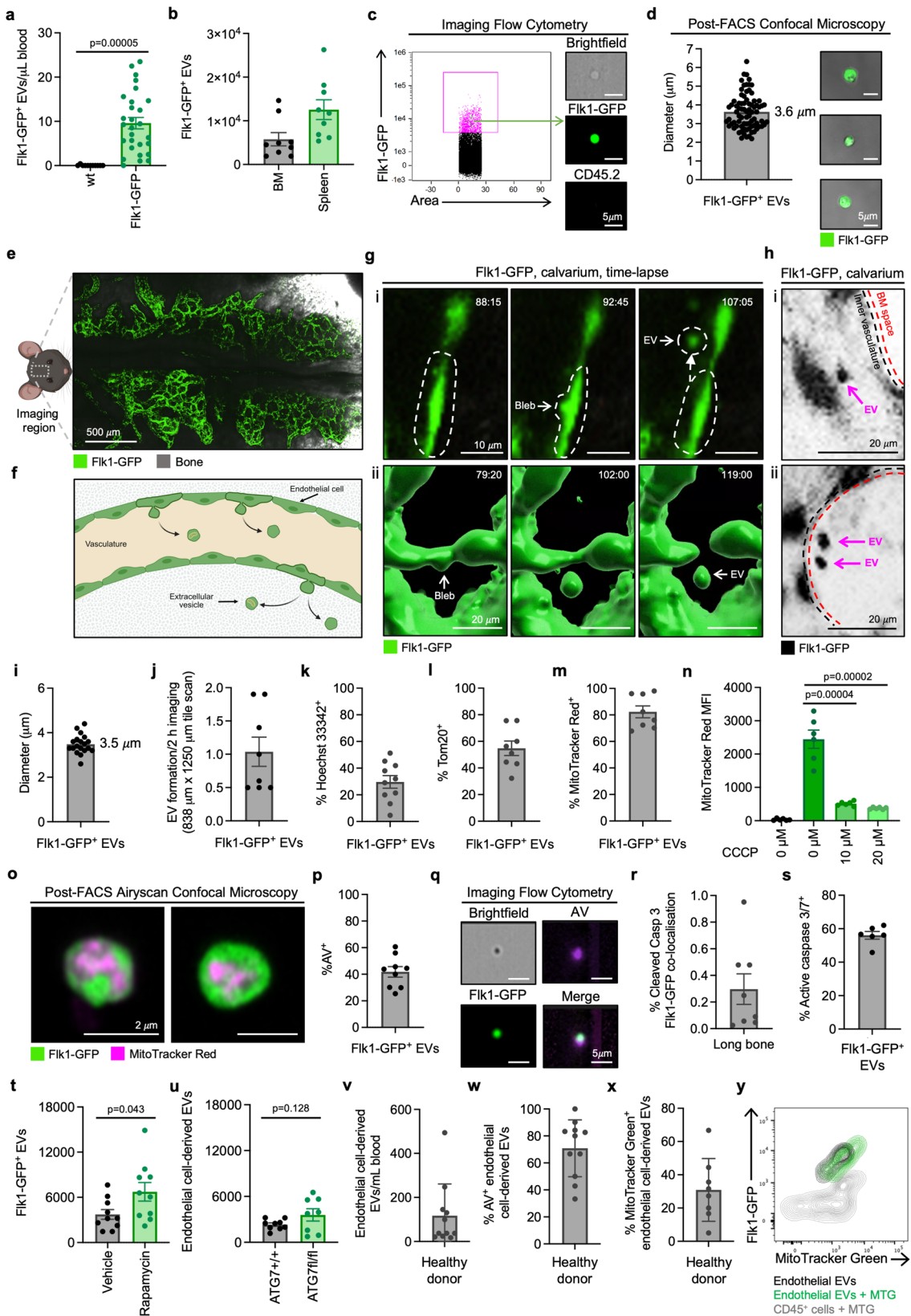

autophagic cells, we determined the level of homoeostatic endothelial cell apoptosis by staining for cleaved caspase 3 in long bone sections using combined confocal multiphoton microscopy. Consistent with the suggested low turnover rates of endothelial cells[46,47], cleaved caspase 3[+] Flk1-GFP[+] endothelial cells were present in long bones at a very low frequency (Fig. 1r). Similarly, cell division could not be detected in

our 4D time-lapse imaging experiments (Supplementary Fig. 3d; Supplementary Video 9). However, approximately 60% of Flk1-GFP[+] EVs contained active caspase 3/7, suggesting that a proportion of the EV population may be derived from apoptotic endothelial cells (Fig. 1s). To further examine whether EVs were generated from dying cells, we performed time-lapse intravital microscopy of the BM calvarium after

**Fig. 1 | Endothelial cells generate large, mitochondria-rich EVs under homoeostatic conditions. a** Quantification of steady-state levels of circulating GFP⁺ endothelial cell-derived EVs in Flk1-GFP⁺ and wt mice by flow cytometry ($n = 11–28$). **b** Absolute numbers of Flk1-GFP⁺ EVs in the BM (representative of one tibia/femur) and spleen ($n = 9$). **c** Representative imaging flow cytometry analysis of BM-derived Flk1-GFP⁺ EVs. Green = Flk1-GFP, Yellow = CD45.2. **d** Quantification of EV diameter and representative confocal microscopy of FACS-isolated BM-derived Flk1-GFP⁺ EVs. Data points represent individual EVs ($n = 79$). Green = Flk1-GFP. **e** Maximum Intensity Projection (MIP) of intravital microscopy tile scan data showing the imaging area within the BM calvarium of a Flk1-GFP⁺ mouse. Green = Flk1-GFP, Grey = SGH. **f** Schematic diagram of blood vessel architecture in the BM calvarium made with BioRender. **g** Representative time-lapse intravital microscopy data demonstrating endothelial cell blebbing and EV formation: (i) MIP and (ii) 3D rendering. Timestamp presented as min:s. Green = Flk1-GFP. **h** Intravital microscopy illustrating detection of EVs inside the calvarium vasculature (i) or marrow (ii). Black = Flk1-GFP. **i** Quantification of Flk1-GFP⁺ EV diameter from intravital microscopy images. Data points represent individual EVs ($n = 19$). **j** Quantification of EV formation rates during in situ time-lapse imaging. Data points represent the average number of EV formation events observed per 2 h of imaging per mouse ($n = 8$). Quantification of (**k**) DNA⁺ (Hoechst 33342, $n = 10$) and (**l, m**) mitochondria⁺ (Tom20, MitoTracker Red, $n = 8$) BM-derived Flk1-GFP⁺ EVs. **n** Quantification of MitoTracker Red in BM-derived Flk1-GFP⁺ EVs after CCCP treatment. Black bar represents unstained cells, green bars represent MitoTracker Red stained ($n = 6$, two independent repeats). **o** Representative Airyscan confocal microscopy of BM-derived Flk1-GFP⁺ EVs showing MitoTracker Red staining. Green = Flk1-GFP, magenta = MitoTracker Red FM. Quantification of Annexin V (AV) staining of BM-derived Flk1-GFP⁺ EVs by flow cytometry (**p**, $n = 9$) and imaging flow cytometry (**q**). **r** Quantification of cleaved caspase 3⁺ Flk1-GFP⁺ endothelial cells in long bone sections under steady-state conditions captured with dual confocal multiphoton microscopy ($n = 8$). **s** Quantification of caspase 3/7⁺ BM-derived Flk1-GFP⁺ EVs as measured by flow cytometry ($n = 6$). **t** Quantification of BM-derived Flk1-GFP⁺ EVs numbers following Rapamycin (4 mg/kg) or vehicle treatment ($n = 10$). **u** Endothelial cell-derived EV levels in wt (ATG7⁺/⁺;UBCCreERT2cʳᵉ/⁺) and ATG7 floxed (ATG7ᶠˡ/ᶠˡ;UBCCreERT2Cʳᵉ/⁺) mice ($n = 8–9$). Quantification of (**v**) Flk1⁺, CD144⁺, CD31⁺, CD133⁻ endothelial cell-derived EV levels in healthy human peripheral blood ($n = 10$), and the proportion of (**w**) AV⁺ and (**x, y**) MitoTracker Green⁺ EVs ($n = 8–10$). Flk1, CD144 and CD31 are markers for endothelial cell, whereas CD133 is a marker for hematopoietic stem and progenitor cells. Data points represent individual donors. At least three independent experiments were performed for all experiments unless otherwise specified. Data points represent individual mice unless otherwise specified. Error bars represent SEM. Statistical analysis: Unpaired Student's two-tailed t-test, *$p < 0.05$, ***$p < 0.001$.

administration of AV to label the cell death marker PtdSer. Even after the generation of Flk1-GFP⁺ EVs, endothelial cells rarely exhibited AV staining in the imaging window captured (up to 50 min post EV formation) (Supplementary Fig. 5). Notably, only occasionally did EV-forming endothelial cells exhibit apoptotic cell-like morphologies (Supplementary Fig. 4d). Therefore, we next examined the role of autophagy in the formation of Flk1-GFP⁺ EVs by treating mice with the mTOR inhibitor rapamycin[10]. Induction of autophagy through rapamycin treatment was sufficient to increase the number of Flk1-GFP⁺ EVs found in the BM (Fig. 1t). However, genetic deletion of the key autophagic regulator ATG7, did not alter endothelial cell-derived EV (CD45.2⁻, CD31⁺, CD146⁺, Flk1⁺) levels (Fig. 1u). Thus, in comparison to the formation of other large, mitochondria-rich EVs[10], our data suggest that ATG7 may not be a key driver in the formation of similar EV subsets generated by endothelial cells.

Next, we harvested the EV fraction from peripheral blood acquired from healthy donors by using a combination of Ficoll separation and differentiation centrifugation steps. Using this approach, we detected a population of Flk1⁺, CD144⁺, CD31⁺ endothelial cell-derived EVs (Fig. 1v; Supplementary Fig. 6). A proportion of these endothelial cell-derived EVs also exposed PtdSer exposure and contained mitochondria (Fig. 1w–y). Collectively, these data demonstrate that endothelial cells can generate large, mitochondria-containing EVs under homoeostatic settings. As markers for apoptotic bodies and exophers are not well described, EV characteristics such as size, PtdSer exposure, active caspase 3/7 and mitochondria content cannot be used to distinguish these large EV subsets[10,31,48–50]. Thus, in line with the MISEV2023 guidelines[51], we adopted the use of the operational term large endothelial cell-derived EVs to describe EVs being studied herein.

### Endothelial cells generate large EVs within the zebrafish vascular system

Many studies have harnessed lower-order organisms such as the zebrafish (*Danio rerio*) to study EVs in vivo[29,33,52]. Thus, to further support our findings that large EVs are released from the endothelium, we investigated the presence of comparable EV subtypes using a zebrafish model. We used the endothelial cell-specific transgenic zebrafish line Tg(*kdrl*-mCherry), where *kdrl* is a homologue of murine *vegfr2*. Time-lapse confocal microscopy data was collected by imaging zebrafish embryos 72 h post fertilization (hpf) for 3–6 h (Fig. 2a). Consistent with our observations in murine models, time-lapse experiments detected abundant circulating large, endothelial cell-derived EVs within the

embryo vascular network (Fig. 2b; Supplementary Video 10). *Kdrl*-mCherry⁺ EVs were ~2.4 μm in diameter and could also be detected by flow cytometry on dissociated embryos (Fig. 2c, d; Supplementary Fig. 7). The characteristics of these EVs were similar to Flk1-GFP⁺ EVs found in mice whereby ~40% of *kdrl*-mCherry⁺ EVs exposed PtdSer and ~80% harboured mitochondria content (Fig. 2e, f). As ~50% of *kdrl*-mCherry⁺ EVs contained active caspase 3/7 (Fig. 2g), a proportion of *kdrl*-mCherry⁺ EVs may be derived from apoptotic endothelial cells. To determine if a proportion of these EVs were also generated through an autophagy-dependent mechanism, we treated *kdrl*-mCherry embryos with the autophagic inhibitor chloroquine (CQ) for 14 h as previously described[53] and characterized EVs via confocal microscopy and flow cytometry. Time-lapse microscopy analysis indicated less circulating *kdrl*-mCherry⁺ EVs in the blood vessels of CQ-treated embryos compared to untreated controls (Fig. 2h). Moreover, flow cytometry of dissociated embryos confirmed that CQ treatment could reduce the total number of *kdrl*-mCherry⁺ EVs (Fig. 2i). Thus, our data from two different model organisms and human samples suggest that production of large, heterogenous mitochondria-rich EVs from endothelial cells under homoeostatic conditions may be a highly conserved process.

### Diverse immune cell subsets interact with, and engulf, endothelial cell-derived EVs

Phagocytic removal of cellular waste is an essential homoeostatic process. When this process is impaired, the accumulation of cellular debris and large EVs leads to inflammation and tissue damage[10,54,55]. As endothelial cell-derived EVs generated under homoeostatic settings can harbour nuclear- and mitochondrial-derived DAMPs (Fig. 1k–n), we investigated whether immune cell subsets could interact with these EVs as a potential clearance mechanism. To first determine whether Flk1-GFP⁺ EVs could be engulfed by professional phagocytes like macrophages, we performed an in vitro engulfment assay. BM-derived Flk1-GFP⁺ EVs were isolated and labelled with the pH-sensitive dye CypHer-5E that fluoresces under acidic conditions (such as in the phagolysosome) and incubated with J774 murine macrophages. Approximately 16 h post-co-incubation, CypHer-5E⁺ J774 macrophages could be observed by confocal microscopy (Fig. 3a), indicating that phagocytes can recognise and engulf large endothelial cell-derived EVs. We next investigated if Flk1-GFP⁺ EVs could interact with and/or be engulfed by immune cells in vivo. Using flow cytometry, we identified a population of CD45.2⁺ immune cells with GFP fluorescence in the blood, BM and spleen (Fig. 3b, c). As GFP expression is limited to

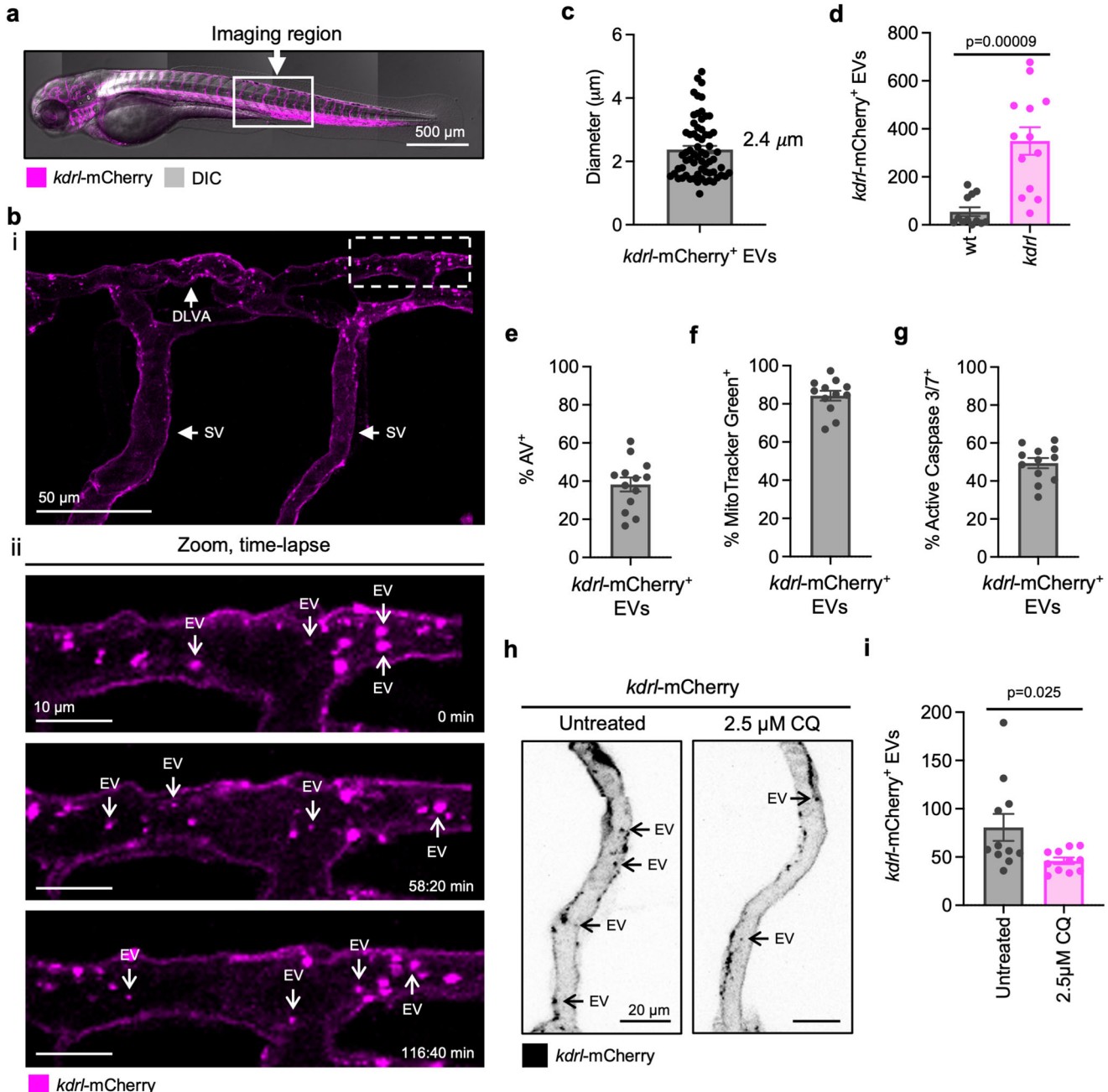

**Fig. 2 | Zebrafish generate large, endothelial cell-derived extracellular vesicles in vivo. a** Confocal microscopy tile scan of a *kdrl*-mCherry zebrafish embryo at 3 dpf highlighting the imaging area for time-lapse experiments. Magenta = kdrl-mcherry. **b** Representative images from time-lapse confocal microscopy data of *kdrl*-mCherry embryo shown as a Maximum Intensity Projection (MIP). Magenta = kdrl-mCherry **c** Quantification of *kdrl*-mCherry⁺ EV diameter from confocal microscopy experiments. Data acquired from *n* = 3 individual embryos, dots represent individual EVs (*n* = 65). **d** Quantification of the total number of *kdrl*-mCherry⁺ EVs per pool of dissociated wt or *kdrl*-mCherry zebrafish embryos. Each data point represents a pool of 10 embryos (*n* = 12–13). Quantification of Annexin V⁺ (AV) (**e**, *n* = 13), MitoTracker Green⁺ (**f**, *n* = 12)) and active caspase 3/7⁺ (**g**, *n* = 12) *kdrl*-mCherry⁺ EVs by flow cytometry. *kdrl*-mCherry embryos at 57 hpf were treated with CQ for 14 h and EVs were assessed by confocal microscopy (**h**, black = kdrl-mCherry) and flow cytometry (**i**). Data points represent a pool of 10 embryos (*n* = 11). Error bars represent SEM. Statistical analysis: Unpaired Student's two-tailed *t*-test, *p < 0.05, ***p < 0.001. At least three independent experiments were performed for all experiments. DLVA dorsal longitudinal anastomotic vessel; SV segmental vein.

*vegfr2*-expressing endothelial cells, we hypothesized that CD45.2⁺ GFP⁺ cells may represent immune cells that were either interacting with or had engulfed, endothelial cell-derived EVs. Broad immune cell phenotyping by flow cytometry indicated that multiple immune cell populations including neutrophils, macrophages, monocytes, dendritic cells, T cells and B cells exhibited GFP fluorescence (Fig. 3d, e). Confocal microscopy performed on FACS-isolated GFP⁺ CD45.2⁺ cells from Flk1-GFP BM confirmed that distinct GFP⁺ EVs could be observed

at the cell periphery (external) as well as within the immune cell (internal) (Fig. 3f, g). Imaging flow cytometry also confirmed that a proportion of GFP⁺ immune cell subsets such as Ly6G⁺ neutrophils and Ly6C⁺ monocytes contained GFP⁺ EVs internally and attached to the cell periphery (Fig. 3h). Thus, multiple immune cell populations can interact with and internalise Flk1-GFP⁺ EVs.

Cellular debris and EVs can display 'eat-me' signals such as PtdSer to facilitate interactions with phagocytes. Flow cytometry analysis

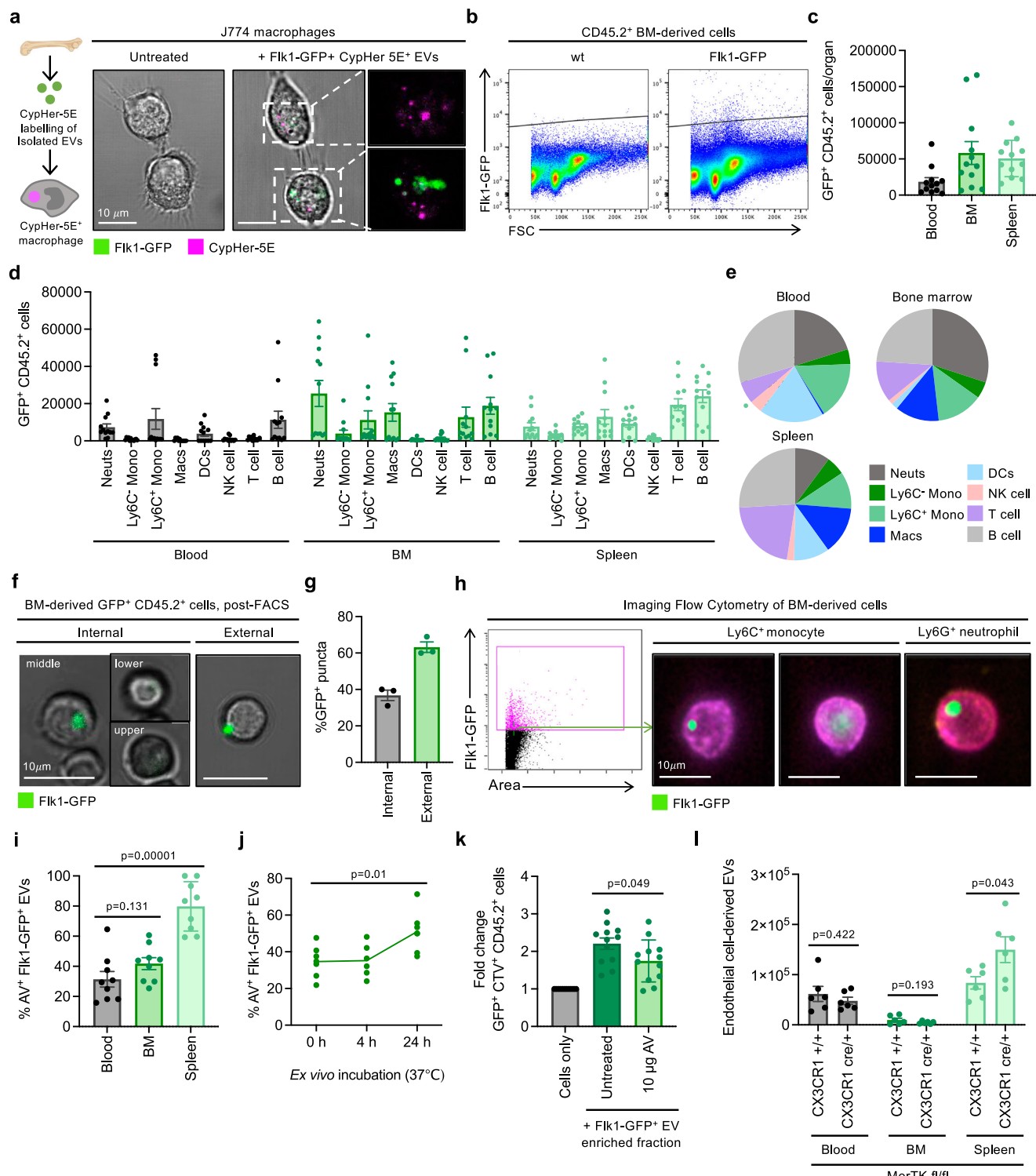

demonstrated that ~30–40% of Flk1-GFP⁺ EVs in the blood and BM, and approximately 80% in the spleen, exhibited PtdSer exposure (Fig. 3i). Interestingly, the proportion of AV⁺ Flk1-GFP⁺ EVs increased when samples were incubated over time ex vivo (Fig. 3j). Thus, after release, it is possible that EVs may gradually expose PtdSer to aid phagocytic recognition. To determine if PtdSer exposure on EVs may contribute to immune cell recognition, BM-derived Flk1-GFP⁺ EV enriched samples were treated with recombinant AV to mask PtdSer, and their interaction with CTV-labelled BM cells derived from wt mice was assessed ex vivo. Notably, co-incubation of the Flk1-GFP⁺ EV enriched sample with CTV-labelled BM cells for 1 h resulted in a significant increase in

the number of GFP⁺ immune cells (Fig. 3k), further supporting our data that immune cells can interact with endothelial-cell derived EVs. Moreover, AV treatment could reduce the number of GFP⁺ immune cells (Fig. 3k). PtdSer is recognised by a broad repertoire of engulfment receptors and bridging molecules. This includes the bridging molecules Gas6/Protein S that recognize exposed PtdSer and bind MerTK on immune cell surfaces[56]. Notably, MerTK has a vital role in the removal of apoptotic materials and exophers[10,57,58]. Thus, we adopted a genetic model to deplete MerTK in CX3CR1-immune cells (MerTK^fl/fl CX3CR1^cre/+) and examined whether this could impact the levels of endothelial cell-derived EVs. Consistent with the reported role of

**Fig. 3 | Endothelial cell-derived EVs interact with and are engulfed by immune cells. a** Representative confocal microscopy images demonstrating uptake of CypHer-5E labelled BM-derived Flk1-GFP⁺ EVs by J774 macrophages. Green = Flk1-GFP, magenta = CypHer-5E. **b** Representative flow cytometry data monitoring GFP fluorescence of CD45.2⁺ immune cells in bone marrow (BM) from wt and Flk1-GFP mice. Quantification of total GFP⁺ CD45.2⁺ immune cells (**c**) or immune cell subsets (**d, e**) in the blood, BM or spleen of Flk1-GFP mice (*n* = 12). BM data is representative of one tibia/femur. Dark grey = neutrophils, light grey = B cells, dark green = Ly6C⁻ monocytes, light green = Ly6C⁺ monocytes, dark blue = macrophages, light blue = dendritic cells, pink = natural killer cells, purple = T cells. **f** Confocal microscopy of BM-derived GFP⁺ CD45.2⁺ immune cells isolated by fluorescence-activated cell sorting. Green = Flk1-GFP. **g** Quantification of internal and external (peripheral) GFP⁺ puncta of confocal microscopy data in (**f**) (*n* = 3).

**h** Representative imaging flow cytometry data of BM-derived GFP⁺ neutrophils (Ly6G⁺) and monocytes (Ly6C⁺). Green = Flk1-GFP, purple = Ly6C, pink = Ly6G. **i** Quantification of Annexin V (AV) binding to Flk1-GFP⁺ EVs in the blood, BM and spleen (*n* = 9). **j** Time course quantification of AV binding to blood-derived Flk1-GFP⁺ EVs during ex vivo incubation at 37°C (n=6−7). **k** Fold change in GFP⁺ CD45.2⁺ immune cells after incubation of wt BM with supernatant from Flk1-GFP BM in the context of recombinant AV blocking (*n* = 12). **l** Quantification of endothelial cell-derived EVs (Flk1⁺, CD31⁺, CD146⁻, CD45.2⁻, FSCˡᵒʷ) in the blood, BM or spleen of MerTKᶠˡ/ᶠˡ Cx3CR1⁺/⁺ or MerTKᶠˡ/ᶠˡ Cx3CR1ᶜʳᵉ/⁺ mice (*n* = 6, two independent repeats). Error bars represent SEM. Statistical analysis: Unpaired Student's two-tailed *t*-test, ns = *p* > 0.05, *\*p* < 0.05, \*\**p* < 0.01, \*\*\**p* < 0.001. At least three independent repeats were performed for all experiments unless otherwise specified. Data points represent individual mice.

MerTK in exopher clearance[10], MerTKᶠˡ/ᶠˡCX3CR1ᶜʳᵉ/⁺ mice exhibited an accumulation of endothelial cell-derived EVs in the spleen, compared to MerTKᶠˡ/ᶠˡCX3CR1⁺/⁺ controls (Fig. 3l). Collectively, these data demonstrate that large, endothelial cell-derived EVs may interact with, and/or be engulfed by various immune cell populations, and this can be facilitated through a PtdSer-dependent mechanism.

## Endothelial cell stress induced by AML results in enhanced formation of endothelial cell-derived EVs

Our data demonstrate that endothelial cells generate large EVs that interact with distinct immune cell subsets under steady state conditions. Therefore, we investigated how this homoeostatic system responded in the context of direct endothelial cell stress. To do so, we induced intense inflammatory insult using a transplantable model of the aggressive blood cancer acute myeloid leukaemia (AML). We chose AML as blood vessel remodelling in the BM has been extensively described in both human disease and mouse models due to the distinct pro-inflammatory profile of this malignancy[43]. We used clones derived from AML cells that express the oncogene MLL-AF9 that recapitulates a commonly observed chromosomal translocation observed in AML patients[59]. We used MLL-AF9 cells (herein referred to as AML) generated from primary mTmG BM cells that constitutively express membrane targeted tdTomato enabling them to be traced in flow cytometry and imaging experiments concurrently with Flk1-GFP[43,60]. We first examined whether the endothelial cell stress caused by AML altered the levels of circulating endothelial cell-derived EVs. Flk1-GFP mice were intravenously injected with AML cells and circulating levels of Flk1-GFP⁺ EVs were assessed by flow cytometry on day 18 post-transplantation when AML has infiltrated the blood and organs such as the BM and spleen[43,61,62]. Strikingly, we observed a significant increase in the number of circulating Flk1-GFP⁺ EVs in the blood of AML-burdened mice (Fig. 4a). Importantly, the number of circulating Flk1-GFP⁺ EVs correlated to the progressive proliferation of AML cells in the blood (Fig. 4b). Expansion of AML also resulted in elevated levels of Flk1-GFP⁺ EVs in the spleen, whilst the overall level of Flk1-GFP⁺ EVs found in the BM did not change (Fig. 4c). Flk1-GFP⁺ EVs derived under AML conditions also displayed properties such as PtdSer exposure and active caspase 3/7 content however, contained less mitochondria (Supplementary Fig. 8). Next, we explored whether the increase in Flk1-GFP⁺ EVs was a result of AML-induced endothelial cell remodelling. We performed multiday intravital microscopy of the BM calvarium and tracked how the BM vasculature responds to AML in situ. AML infiltration induced major remodelling of Flk1-GFP⁺ blood vessels during late-stage disease and resulted in the degradation of entire Flk1-GFP⁺ blood vessels in the BM calvarium (Fig. 4d). Imaging of both the calvarium and long bones at ethical endpoint further highlighted the significant degradation of GFP⁺ blood vessels observed in AML-burdened mice, as determined by the quantification of total Flk1-GFP⁺ pixel count and vessel density (Fig. 4e−g). To determine if endothelial cell loss was driven by cell death mechanisms, we harvested long bones and stained bone sections for the presence of

cleaved caspase 3. Dual confocal multiphoton analysis demonstrated a modest increase in the co-localisation of cleaved caspase 3 and Flk1-GFP⁺ endothelial cells in mice with AML (Fig. 4h, i). Furthermore, AV⁺ Flk1-GFP⁺ cells and fragments could also be observed in the BM microenvironment of AML-burdened mice by intravital microscopy of the BM calvarium, indicating that endothelial cells were undergoing cell death (Fig. 4j; Supplementary Fig. 9; Supplementary Video 11). Together, these data demonstrate that AML can enhance the number of Flk1-GFP⁺ EVs, and this is likely a result of vasculature degradation caused by late-stage AML.

To examine whether the enhanced formation of Flk1-GFP⁺ EVs observed during AML was a direct result of increased EV production, we performed a series of in vivo microscopy studies on the BM calvarium in AML-burdened mice at day 18 post-transplantation (Fig. 5a; Supplementary Video 12). Intravital microscopy of individual blood vessels (acquiring images every 1 s for 5 min), demonstrated an accumulation of Flk1-GFP⁺ EVs in the vasculature of mice with AML, compared to steady-state controls (Fig. 5b). Excitingly, time-lapse studies of the BM calvarium captured the formation of Flk1-GFP⁺ EVs in AML-burdened mice, where EVs were released as a result of direct budding from the endothelium or occasionally, by complete endothelial cell fragmentation (Fig. 5c, e, f; Supplementary Videos 13−14). The generation of Flk1-GFP⁺ EVs in AML-burdened mice was observed at a rate of ~2.7 events/2 h of imaging (Fig. 5c), substantially more frequently than the release of Flk1-GFP⁺ EVs observed under steady-state conditions (Fig. 1j). Flk1-GFP⁺ EVs observed in mice with AML were also defined as large EVs, given their approximate diameter of 3.4 μm (Fig. 5d), consistent with those observed in steady-state mice (Fig. 1i). Taken together, our data has revealed that endothelial cells can generate large EVs under both homoeostatic and malignant settings, and that endothelial cell stress and death caused by AML can directly increase the generation of Flk1-GFP⁺ EVs in vivo.

Given that AML could enhance the levels of Flk1-GFP⁺ EVs, we also investigated the systemic interactions of such EVs with the immune cell populations present during AML. The expansion of AML in vivo results in a significant change in the proportion of immune cell subsets, such as an increase in monocytes, neutrophils, NK cells, T cells and B cells in the blood, and depletion of neutrophils, monocytes and B cell populations in the BM (Supplementary Fig. 10a). As such, the presence of AML altered the proportion of GFP⁺ immune cell populations (i.e. immune cells likely interacting with or engulfing Flk1-GFP⁺ EVs). This included a significant decrease in the percentage of GFP⁺ neutrophils in the BM, GFP⁺ macrophages and B cells in the spleen, and an increase in the absolute number as well as the percentage of GFP⁺ NK cells in the blood (Supplementary Fig. 10b, c).

## Increased levels of circulating endothelial cell-derived EVs may provide a general insight into vasculature health during blood cancer

To determine if the increase in large, endothelial cell-derived EV levels observed during AML is disease-specific or a result of general BM

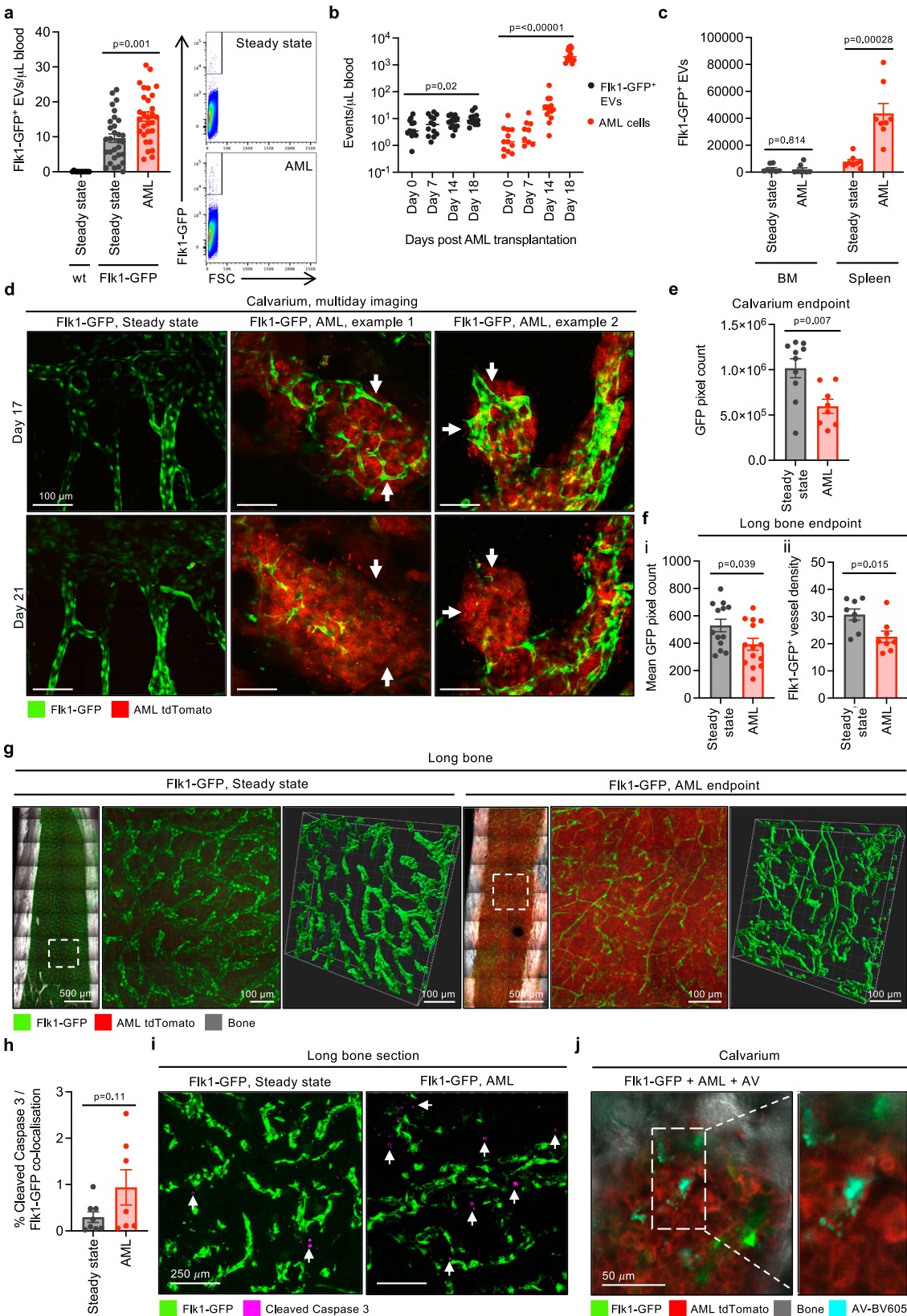

infiltration, we applied our approach to two additional models of blood malignancies derived from distinct cell lineages. We used the EµMyc 560 model that recapitulates the *immunoglobulin/c*-MYC chromosomal translocation observed in aggressive lymphomas such as Burkitt Lymphoma[63]. We also used an intracellular Notch-driven model of T cell acute lymphoblastic leukaemia (herein referred to as T-

ALL) that expresses cytoplasmic dsRed and mimics aggressive T-ALL observed in humans[42]. The infiltration of EµMyc 560 cells did not alter the number of circulating Flk1-GFP+ EVs observed at ethical endpoint (Fig. 6a). However, similar to AML, the expansion of T-ALL resulted in a significant increase in the levels of circulating Flk1-GFP+ EVs detected at day 18 post-transplantation when leukaemic cells had infiltrated the

**Fig. 4 | AML elevates levels of circulating endothelial cell-derived EVs at late-stage disease.** Quantification of circulating Flk1-GFP⁺ EVs in steady state and AML-burdened mice at (**a**) 18–19 days post-transplantation (DPT, $n = 12$–30) and (**b**) at regular intervals following transplantation ($n = 12$–13). **c** Absolute number of Flk1-GFP⁺ EVs in the BM and spleen of steady-state or AML-burdened mice, 18 DPT ($n = 8$). BM data is representative of one tibia/femur. **d** Maximum intensity projection (MIP) of the calvarium endothelium in steady state or AML-burdened mice (17 and 21 DPT) showing endothelial cell loss. Green = Flk1-GFP, red = MLL-AF9 tdTomato. **e** Quantification of the total GFP pixel count of intravital microscopy data of the calvarium in steady state or AML-burdened mice, harvested at ethical endpoint ($n = 8$–10). **f** Quantification of dual confocal multiphoton microscopy of long bones harvested from steady-state or AML-burdened mice at ethical endpoint

showing the mean GFP pixel count (i, $n = 13$–14) and vessel density (ii, $n = 8$). **g** Representative MIPs. Green = Flk1-GFP, red = MLL-AF9 tdTomato, grey = SGH. **h, i** NLO dual confocal multiphoton microscopy of cleaved caspase 3⁺ Flk1-GFP⁺ cells in steady state and AML-burdened long bone sections (18 DPT, $n = 7$–8). Images shown as MIP. Steady-state control data also shown in Fig. 1r. Green = Flk1-GFP, magenta = cleaved caspase 3. **j** Intravital microscopy of Annexin V⁺ (AV) endothelial cells and fragments in the calvarium of AML-burdened mice (18–21 DPT). Green = Flk1-GFP, red = MLL-AF9 tdTomato, grey = SGH, cyan = AV-BV605. Error bars represent SEM. Statistical analysis: Unpaired Student's two-tailed t-test, ns = $p > 0.05$, *$p < 0.05$, **$p < 0.01$, ***$p < 0.001$. At least three independent repeats were performed for all experiments. Data points represent individual mice unless otherwise specified.

BM and blood (Fig. 6b). Importantly, the levels of circulating Flk1-GFP⁺ EVs observed in both models correlated to the contrasting levels of endothelial cell loss (Fig. 6c–j). For example, although EµMyc 560 cell infiltration induced endothelial cell remodelling in the BM calvarium and long bones, the presence of EµMyc 560 did not result in a loss of Flk1-GFP⁺ blood vessels at ethical endpoint (Fig. 6c, d, g, i). In contrast, T-ALL induced a marked reduction in Flk1-GFP⁺ blood vessels observed in both the calvarium and long bones at ethical endpoint, indicating that T-ALL could result in the degradation of the endothelium at late-stage disease (Fig. 6e, f, h, j). Collectively, by examining multiple models of blood malignancies, our data demonstrate that increased levels of circulating large endothelial cell-derived EVs may provide a general insight into the extent of endothelium degradation during the progression of blood cancer in a lineage-specific manner.

## Discussion

EV research has rapidly evolved in recent years and has demonstrated that EV formation is a fundamental mechanism of intercellular communication. Although the visualization of EV production under in vitro settings[64–66] or in lower-order organisms such as *D. rerio*[33], *C. elegans*[67] and *Drosophila*[68] is well described, complementary imaging studies in mouse models are lacking[69]. Here, we performed comprehensive intravital imaging of endothelial-reporter mice and characterized the generation of large, endothelial cell-derived EVs within the BM calvarium of mice under homoeostatic and malignant conditions. Thus, our study provides visualization and spatio-temporal characterization of endothelial cell-derived EV production in a murine model. In particular, we determined the frequency of large endothelial cell-derived EV formation in situ and their levels in various organs, which are fundamental understandings that cannot be determined from in vitro systems. These technical advances are significant to the EV and cell biology field as they provide direct evidence of EV formation, enabling the discovery of important in vivo functions. For example, direct visualization of the formation and clearance of apoptotic bodies has revealed the intricate interaction between fragmenting apoptotic cells and phagocytes[35,37,70], as well as the role of this process in the regulation of cell proliferation through the trafficking of signalling molecules like Wnt8a[34]. In vivo imaging studies have also captured the formation of migrasomes in zebrafish[29] and mice[30], which complemented landmark discoveries detailing how migrasomes can aid organ morphogenesis[29] and mitochondrial quality control[30]. Moreover, imaging approaches have been vital in validating the mechanism of exosome biogenesis in vivo, exosome trafficking, and uncovering a role of exosomes in aiding vascular development[33]. Collectively, the development of novel imaging techniques that are equipped to examine EVs in vivo can facilitate major advances in cell biology. Thus, the data presented here are fundamental for understanding the formation and function of large endothelial cell-derived EVs.

Mitochondria-containing EVs can significantly impact the surrounding tissue microenvironment through their roles in inflammation, intercellular communication, mitochondria quality control and waste disposal[10,30,71,72]. Therefore, understanding the formation

of EV subsets that contain mitochondria is of broad interest in both steady state and disease settings. By harnessing two distinct model organism systems and analysing human peripheral blood, we identified a population of large, mitochondria-containing EVs generated by endothelial cells under homoeostatic conditions. In line with the MISEV 2018 & 2023 guidelines[8,51], we categorized these endothelial cell-derived EVs into a general subgroup of 'large EVs' to encompass the heterogeneous nature of EVs generated in vivo. As these large endothelial cell-derived EVs are clearly more than 1 µm in diameter, they are distinct from submicron, small EV subsets such as exosomes and microvesicles. Notably, the described large endothelial cell-derived EVs likely represent a mixture of different EV subsets, including but not limited to, apoptotic body-like EVs (based on PtdSer exposure and active caspase 3/7 content) and exopher-like EVs (mitochondria-rich and generated from viable cells). Although other large EVs subsets such as migrasomes can also contain mitochondria under conditions of cellular stress[30], the large endothelial cell-derived EVs as described in this study are unlikely to represent migrasomes as they do not appear to be generated from the trailing edge of migrating endothelial cells. Furthermore, the presence of mitochondria in the majority of large endothelial cell-derived EVs is unexpected as endothelial cells possess relatively low mitochondria content (~2–6%) in comparison to other cell types that generate mitochondria-rich EVs like cardiomyocytes (~30%)[10,73,74]. Thus, whether endothelial cells can specifically package mitochondria into large EVs to aid communication with cells at distal sites would be of interest to explore. Notably, we observed that large endothelial cell-derived EVs may not expose the 'eat-me' signal PtdSer immediately after their formation but rather, gradually expose PtdSer overtime. This may represent a mechanism whereby these large EVs are not cleared at the site of their formation but, are removed by specialised phagocytes at distal organs when a sufficient amount of PtdSer is exposed. This observation is consistent with our findings that a greater proportion of large endothelial cell-derived EVs in the spleen expose PtdSer, and EVs accumulate in the spleen when mice lack the efferocytic receptor MerTK. As the rapid clearance of other large, mitochondria-containing EVs is essential in maintaining tissue homoeostasis[10], clearance of large endothelial cell-derived EVs by different immune cell subsets may serve a similar role to limit the release of intracellular DAMPs. Notably, the accumulation and subsequent rupture of mitochondria-rich endothelial cell-derived EVs could also provide a potential answer for the elevated levels of cell-free mtDNA observed in numerous inflammatory disease pathologies[75]. Inflammatory diseases such as diabetes mellitus and atherosclerosis are linked with altered mitochondria dynamics that can cause endothelial cell dysfunction[73,76] and exhibit enhanced endothelial cell apoptosis[77–79]. Here, our study has focused on the visualisation of large EV production in situ and their subsequent interaction with immune cells under homoeostatic conditions. However, fundamental questions such as how inflammation regulates the generation of large endothelial cell-derived EVs, the molecular mechanism(s) of their biogenesis and their functional

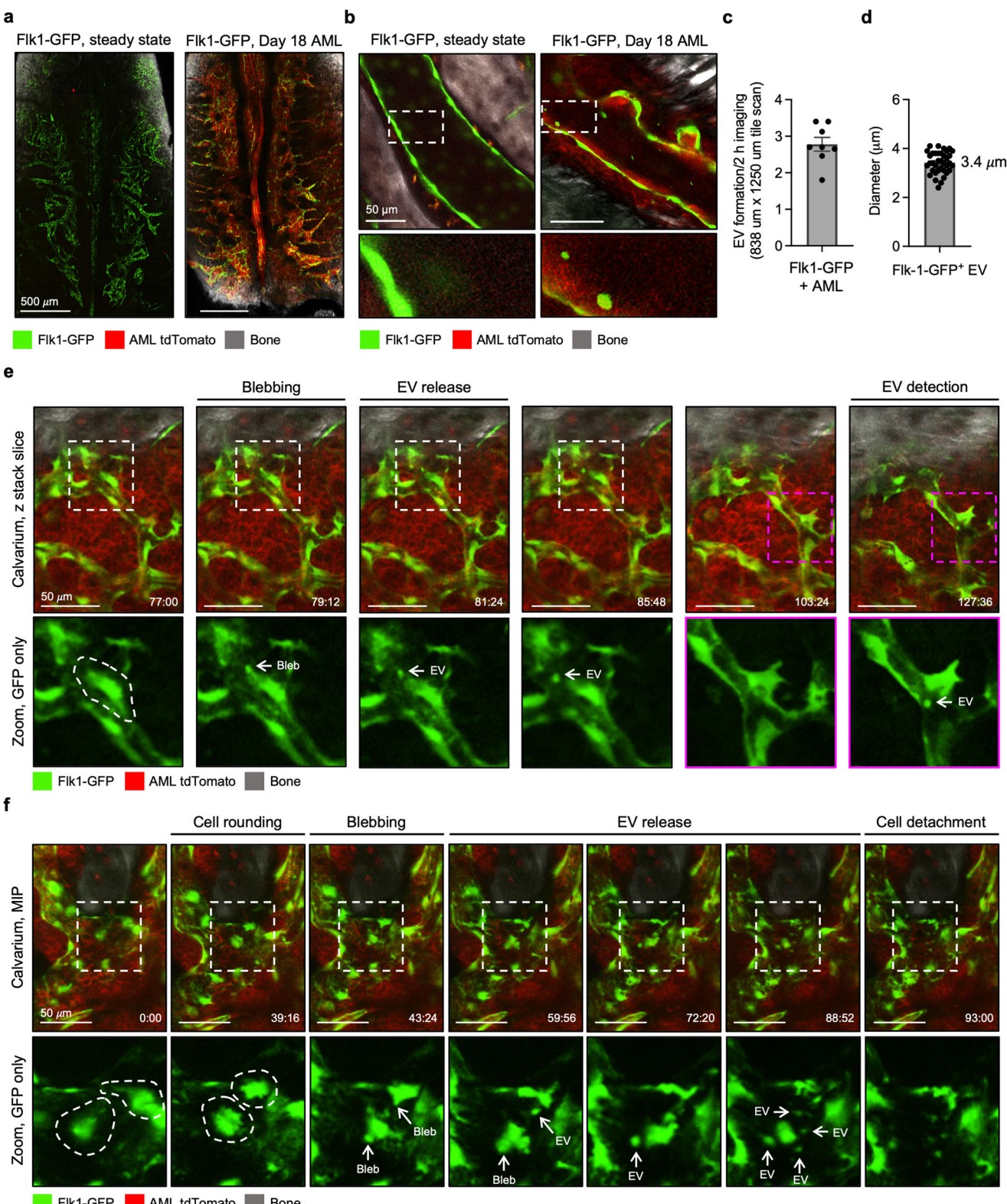

**Fig. 5 | In vivo imaging of endothelial cell EV formation and cell fragmentation during AML. a** Representative calvarium Maximum Intensity Projection (MIP) of steady-state or AML-burdened Flk1-GFP mice. **b** MIP of time-lapse intravital microscopy analysis performed on individual blood vessels in steady state or AML-burdened mice (projection of merged images taken every 1 s for 5 min). **c** Quantification of EV formation by intravital microscopy of AML-burdened mice. Each data point represents the average number of EV formation events observed per 2 h of imaging per mouse ($n = 8$). **d** Quantification of Flk1-GFP⁺ EV diameter

derived from intravital microscopy data. Each data point represents an individual EV ($n = 37$). **e, f** Example of Flk1-GFP⁺ cell blebbing and EV formation in AML-burdened mice. Timestamp presented as min:s. Error bars represent SEM. Statistical analysis: Unpaired Student's two-tailed $t$-test, ns = $p > 0.05$, *$p < 0.05$, ***$p < 0.001$. At least three independent experiments were performed for all experiments. AML-burdened mice assessed at 18 days post-transplantation (DPT). For all microscopy images shown, green = Flk1-GFP, red = MLL-AF9 tdTomato and grey = SGH.

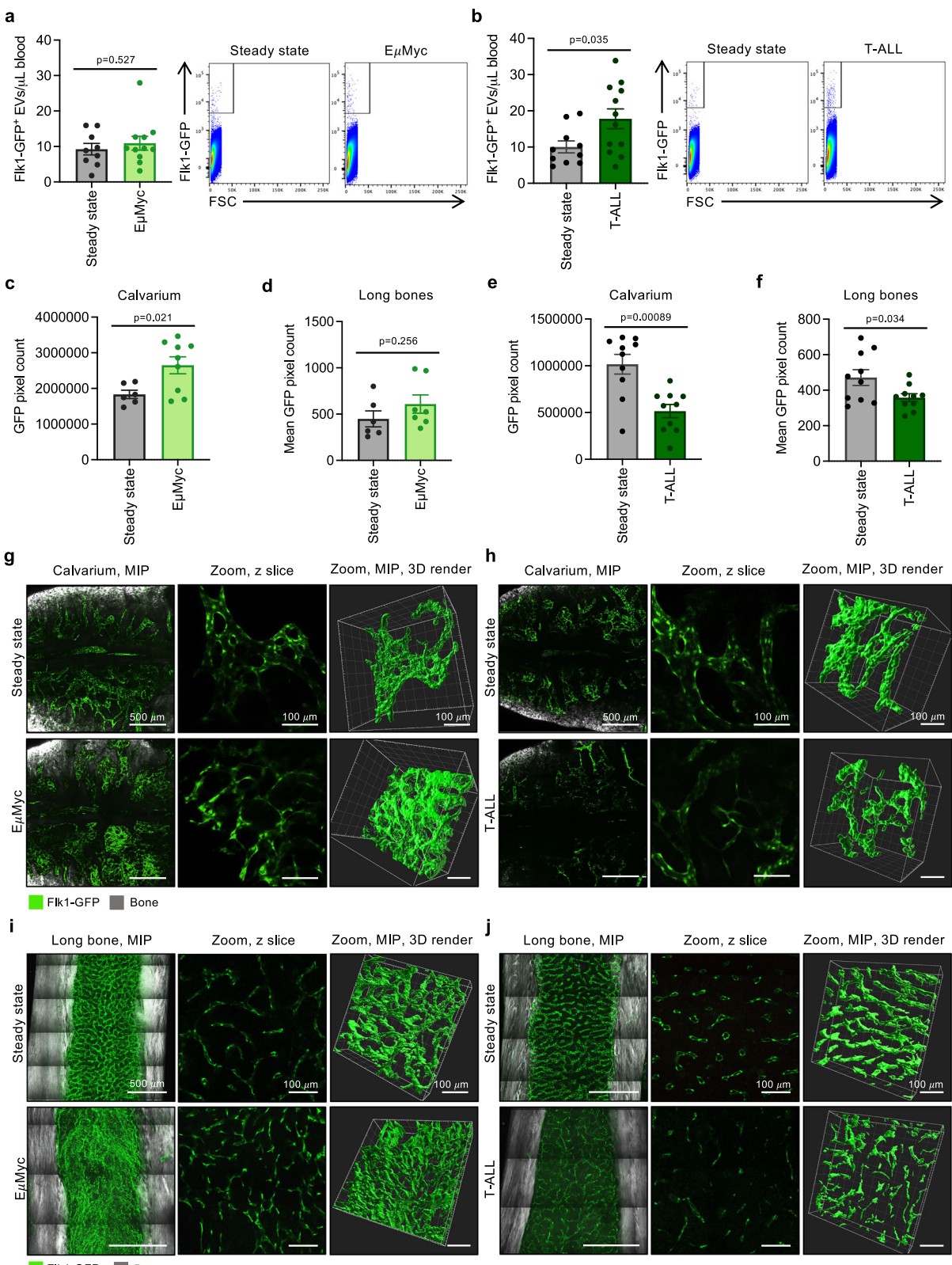

impact on recipient cells remain to be fully defined and would be of significant interest in future studies.

Aberrant inflammatory signalling is strongly associated with blood malignancies and is often linked with disease pathogenesis[80]. For example, AML patients exhibit elevated levels of inflammatory cytokines such as IL-1β, IL-6, IL-8, IL-10 and TNF[81–84], where IL-1β can aid

disease progression by promoting AML cell growth[83]. Similarly, IL-6 may also support disease pathogenesis as AML patients with low IL-6 levels are associated with increased survival[81,82]. Notably, inflammatory profiles are further altered in patients who receive chemotherapy[85]. It is therefore vital to understand the role of inflammation in the tissue microenvironment during the onset, progression and treatment of

**Fig. 6 | Circulating endothelial cell-derived EV numbers correlate with endothelial cell loss in blood malignancies.** Flk1-GFP mice were transplanted with either (**a**) EμMyc B cell lymphoma or (**b**) dsRed T cell leukaemia (T-ALL) that over-expresses Intracellular Notch (ICN) and circulating levels of Flk1-GFP⁺ EVs quantified by flow cytometry on 14 and 18 days post-transplantation (DPT), respectively (*n* = 9–12). Steady state control data also shown in Fig. 4a. **c–f** Quantification of GFP⁺ pixels in the calvarium and long bones from EμMyc and T-ALL burdened mice at

ethical endpoint (**g**, **h**) (*n* = 6–10). Representative images of the calvarium (**g**, **h**) and long bones (**i**, **j**). Data are shown as Maximum Intensity Projection (MIP) of whole organ, a zoomed section and 3D render for each condition. Error bars represent SEM. Statistical analysis: Unpaired Student's two-tailed *t*-test, ns = *p* > 0.05, \**p* < 0.05, \*\*\**p* < 0.001. At least three independent repeats were performed for all experiments unless otherwise specified. Data points represent individual mice unless otherwise specified. For all microscopy images shown, green = Flk1-GFP and grey = SGH.

blood cancers. Here, we discovered that the MLL-AF9-driven model of AML results in enhanced release of endothelial cell-derived EVs, which correlated with endothelial cell loss during late-stage disease. Considering such a significant loss of the BM vasculature[43], it is surprising that circulating levels of large, endothelial cell-derived EVs only increased by approximately two-fold. This finding further supports the concept that large EVs are removed by a highly efficient homoeostatic clearance mechanism. Similar results were also observed when adopting a transplantable model of T-ALL where increased circulating EVs also correlated with endothelial cell damage. However, increased circulating levels of endothelial cell-derived EVs were not detected in a model of B cell lymphoma where we observed significant endothelial cell remodelling, but not degradation. This highlights that increased EV formation is a result of altered endothelial cell behaviour (likely death) caused by distinct lineages of malignancies, rather than simply resulting from mass bone marrow infiltration. Thus, future studies investigating if specific inflammatory profiles regulate the quantity and quality of EVs would be of fundamental importance to the cell biology that underpins this process. Moreover, whether altered immune cell numbers or efferocytosis contributes to the inflammatory profiles observed during malignancies such as AML would also be a significant biological insight. Together, our data suggest that large endothelial cell-derived EVs may constitute an attractive diagnostic tool for the detection of vascular damage via the sampling of peripheral blood. Consistent with this hypothesis, small EVs generated from endothelial cells have previously been investigated in the context of cardiovascular disease[14,77,86–89], viral infection[90] and tumour progression[25]. Thus, understanding the formation and function of different endothelial cell-derived EV subsets, such as the large endothelial cell-derived EVs described here, has wide-spread implications across multiple inflammatory disease settings.

## Methods

### Mice

All mouse experiments were performed in accordance with the WEHI animal ethics committee regulations (AEC 2021.007) and the La Trobe University animal ethics committee (AECs 18024, 18036) in accordance with the Australian code for the care and use of animals for scientific purposes. *Vegf-r2* or Foetal liver kinase 1 (Flk1)-GFP mice (Flk1^GFP/+^)[41] were bred and housed in specific pathogen-free (SPF) conditions at the WEHI animal facility (Parkville, VIC, Australia). Atg7^fl/fl^ (Atg7^tm1Tchi^) mice[91] were provided by the Kile Laboratory (University of Adelaide, SA, Australia). The *UBC-CreERT2* mice[92] were provided by the Heath Laboratory (WEHI, VIC, Australia). Mice were housed at the La Trobe Animal Research and Teaching Facility (LARTF, La Trobe University, VIC, Australia) under SPF conditions. To induce deletion in Atg7^fl/fl^ UBC-CreERT2^+/−^ mice, mice aged 6 weeks or older were intraperitoneally injected with 4 mg tamoxifen (Sigma-Aldrich, #T5648) in sunflower seed oil (Sigma-Aldrich, #25007), delivered as one injection per day of 200 mL of a 10 mg/mL stock, over two consecutive days. Targeted deletion of the *Mertk* gene in Cx3Cr1⁺ cells was achieved by crossing *Mertk*-floxed mice with mice expressing Cre recombinase under the regulatory control of the endogenous *Cx3Cr1* promoter (C57BL/6-Cx3cr1<tm1.1(cre)Jung>/Orl Cx3Cr1cre)[93]. Exon 2 of the *Mertk* gene is flanked by loxP sites and cre-mediated deletion of exon 2 results in an early stop codon. Mice were maintained at the Florey

Institute of Neuroscience and Mental Health, VIC, Australia, in a SPF environment. All mice were maintained on a C57Bl/6 background. Male and female mice at 6–12 weeks of age were used for all experiments where specified (sex-based differences were not considered in this study).

### Cells

J774 macrophages were grown in high glucose (4500 mg/L) Dulbecco's Modified Eagle's Medium (DMEM, Gibco, #11965092) with 10% (v/v) foetal calf serum (FCS), 100 U/mL penicillin and 100 μg/mL streptomycin. Cells were cultured at 37 °C, 5% $CO_2$. EμMyc 560 cells were cultured in 10% (v/v) FCS in high glucose, pyruvate DMEM (Gibco, #11995065), 23.8 mM sodium bicarbonate (Sigma–Aldrich, #S8761), 1 mM HEPES (Gibco, #15630080), 13.5 μM folic acid (Sigma–Aldrich, #F8758), 0.24 mM L-asparagine monohydrate (Sigma–Aldrich, #A7094), 0.55 mM L-arginine mono-hydrochloride (Sigma–Aldrich, #A6969), 22.2 mM D-glucose (Ajax Finechem, #713), 10% (v/v) FSC, 100 U/mL penicillin, 100 μg/mL streptomycin and 50 μM 2-mercaptoethanol (Sigma-Aldrich, #M3148). EμMyc 560 cells were cultured at 37 °C, 10% $CO_2$.

### Blood cancer models

MLL-AF9 membrane tomato membrane GFP (mTmG) acute myeloid leukaemia (AML) cells were generated as previously reported[61,62]. AML was induced by transplanting 200,000 MLL-AF9 mTmG cells intravenously into non-irradiated recipients (approximate endpoint of 19–21 days). T cell acute lymphoblastic leukaemia (T-ALL) was generated as previously reported[42,94]. T-ALL was established by injecting 20,000 intracellular NotchICNΔRamΔP DsRed T-ALL blasts into sublethally irradiated recipient mice (2 × 300 rads with >3 h between doses). For B cell lymphoma models, $1 \times 10^6$ EμMyc 560 B cells were transplanted intravenously into non-irradiated recipient mice (approximate endpoint of 14 days).

### Murine tissue processing

Blood was harvested in EDTA K microtubes and immediately lysed in red blood cell (RBC) lysis buffer for 5 min at room temperature (RT). Samples were topped with 1 mL PBS and centrifuged at 3000 × *g* for 6 min before repeating RBC lysis. The BM samples were prepared by flushing tibias/femurs/hips with PBS using a 1 mL syringe and 25-gauge needle. Samples were filtered through a 70 μm cell strainer and centrifuged at 3000 × *g* for 6 min. For spleen analysis, spleens were harvested and filtered directly through a 70 μm cell strainer into 5 mL of RBC lysis buffer and incubated for 5 min at RT. Samples were topped with PBS and centrifuged at 3000 × *g* for 6 min.

### Fluorescence activated cell sorting (FACS) and flow cytometry of murine samples

To isolate BM-derived EVs via FACS, EVs were first enriched by differential centrifugation. Briefly, multiple flushed BM samples were pooled and centrifuged at 300 × *g* for 10 min to pellet cells. The supernatant was collected and centrifuged at 3000 × *g* for 20 min to pellet EVs. Samples were resuspended in FACS buffer (10% (v/v) FCS, 2 mM EDTA, PBS, 1x Annexin V (AV) binding buffer), stained and filtered through a 70 μm cell strainer prior to sorting. The following flow cytometry gating strategy was used to identify Flk1-GFP⁺ EVs in standard assays: 1) SSC high beads removed; 2) TO-PRO-3 high necrotic cells removed; 3)

CD45.2$^+$ particles removed; 4) FSC low AV low-high particles selected; and 5) GFP high, FSC low endothelial cell-derived EVs selected (Supplementary Fig. 1). An additional gating step was included to remove reporter blood cancer cells when necessary. Flow cytometry analysis was performed on a BD LSRFortessa X-20 or BD FACSCanto II.

The following stains and antibodies were used for flow cytometry and FACS experiments: AV-APC (BD, #550475), -V450 (BD, #560506), -PE (BD, #556421), -PerCPCy5.5 (BD, #559935), -BV605 (BD, #563974) were used at 1:100 in 1x AV binding buffer (BD, #556454) for 10 min at RT; TO-PRO-3 (ThermoFisher, #T3605) was used at 1:1000 for 10 min at RT; MitoTracker Red FM (ThermoFisher, #M22425) was used at 1:4000 for 25 min at 37 °C; Hoechst 33342 (ThermoFisher, #H3570) was used at 1:1000 for 20 min on ice; Active caspase dyes (Vybrant FAM Caspase 3/7 Green: Invitrogen #V35118 and Image-iT™ LIVE Caspase 3/7 Red: Invitrogen #I35102) were incubated at a final concentration of 1x for 1 h at 37 °C; Fc blocking (BD Bioscience, #553142) was performed on ice for 10 min at 1:300 prior to immune cell staining; The following antibodies were incubated for 20 min at 4°C: CD45.2-PerCP Cy5.5 (1:200, BD Bioscience, #552950), -AF700 (1:200, BD Bioscience, #560694), -PE (1:200, BD Bioscience, #560695), -APC Cy7 (1:200, BD Bioscience, #560694), CD11b-500 (1:300, BD Bioscience, #562127), CD11c-APC Cy7 (1:200, BD Bioscience, #561241), Ly6C-BV421 (1:300, BD Bioscience, #562727), Ly6G-PeCy7 (1:300, BD Bioscience, #560601), F4/80-PeCy5 (1:150, eBioscience, #15-4801-82), CD64-BV786 (1:300, BD Bioscience, #741024), I-A/I-E-BV650 (1:300, BD Bioscience, #743873), CD3-AF700 (1:300, BioLegend, #10021), CD4-BV421 (1:300, BD Bioscience, #562891), CD8-BV650 (1:300, BioLegend, #100742), CD19-PeCy7 (1:300, BD Bioscience, #552854), NK1.1-APC Cy7 (1:200, BD Bioscience, #560618), Flk1-PE (1:150, Invitrogen, #12-5821-82), CD31-PeCy7 (1:200, eBioscience, #25-0311-82), CD146-PerCP Cy5.5 (1:200, BD Bioscience, #562231) and CD41-APC Cy7 (1:300, BioLegend, #133928); Tomm20-647 (Abcam, #ab209606) was used at 1:200 for 25 min at RT after samples were fixed in 2% (w/v) paraformaldehyde (PFA) and permeabilized in BD Permeabilization solution.

Immune cell populations were defined by flow cytometry as follows: Neutrophils = CD45.2$^+$, CD11b$^+$, Ly6G$^+$; Macrophages = CD45.2$^+$, CD11b$^+$, Ly6G$^-$, F4/80$^+$, CD64$^+$; Ly6C$^+$ Monocytes = CD45.2$^+$, CD11b$^+$, Ly6G$^-$, F4/80$^-$, Ly6C$^{intermed-high}$; Ly6C$^-$ Monocytes = CD45.2$^+$, CD11b$^+$, Ly6G$^-$, F4/80$^-$, Ly6C$^{low}$; Dendritic cells (DCs) = CD45.2$^+$, CD11b$^-$, CD11c$^+$, MHCII$^+$; B cells = CD45.2$^+$, CD19$^+$; T cells = CD45.2$^+$, CD19$^-$, CD8$^+$/CD4$^+$; Nature Killer (NK) cells = CD45.2$^+$, CD19$^-$, CD8$^-$/CD4$^-$, NK1.1$^+$.

### CCCP treatment
BM from Flk1-GFP$^+$ mice was isolated as above and treated with 0, 10 or 20 μM carbonyl cyanide m-chlorophenyl hydrazone (CCCP; Sigma, #215911) for 10 min at 37 °C. Samples were washed by centrifugation at 3000 × $g$ for 6 min before staining with MitoTracker Red FM, AV V450, TO-PRO-3 and CD45.2 APC Cy7 (as above).

### Human samples
Human peripheral blood samples were acquired from unrelated healthy donors via the Volunteer Blood Donor Registry (WEHI, Parkville, VIC, 3052, Australia). Donors include 3 males (average age 58) and 8 females (average age of 44). Donor age, sex, and medical history were self-reported and sex-based differences were not considered in this study. Informed consent was obtained from all individual participants prior to inclusion in the study. The study was performed according to the principles of the 1964 Helsinki Declaration and its later amendments and was approved by local Human Research Ethics Committee (WEHI Approved project #10/02). Participants were not compensated for their involvement in this study. Whole blood was diluted at 1:1 with PBS and separated by Ficoll gradient. The plasma and mononuclear cell fractions were harvested and centrifuged at 300 × $g$ for 10 min to pellet cells. The supernatant was collected and centrifuged at 3000 × $g$ for 10 min to pellet EVs. The EV fraction was then resuspended in FACS

buffer and stained as follows: AV-BV605 (1:100) and TO-PRO-3 (1:1000) for 10 min at RT; Fc block (1:300) for 10 min on ice; antibody staining with Flk1-PE (1:100, BD Bioscience, 560872), CD31-PeCy7 (1:100, BD Bioscience, 563651), CD144-BV786 (1:200, BD Bioscience, 565672), CD133-BV421 (1:100, BD Bioscience, 566595), CD45-V500 (1:100, BD Bioscience, 560777) for 20 min at 4 °C. Flow cytometry gating was performed as per Supplementary Fig. 6.

### Imaging flow cytometry
BM samples were processed and stained as per 'Methods' section 'Fluorescence Activated Cell Sorting (FACS) and flow cytometry of murine samples' before fixing with 2% (w/v) PFA. Imaging flow cytometry was performed on the Amnis ImageStreamX MkII (Amnis Corporation) using brightfield and fluorescence measurements at 405 nm, 488 nm and 642 nm. Events were captured at 60x magnification with low flow rate/high sensitivity on INSPIRE Software and analysed using IDEAS v6 Software (Amnis Corporation).

### Ex vivo engulfment assay
30,000 J774 macrophages were seeded into a well of an 8-well chamber slide (Ibidi, #80826). The following day, BM-derived Flk1-GFP$^+$ EVs were isolated as described in the 'Methods' section 'Fluorescence activated cell sorting (FACS) and flow cytometry of murine samples' and stained with CypHer-5E NHE Ester (Cytiva, #PA15405) for 40 min at 37 °C. Samples were washed with PBS and EVs were pelleted at 3000 × $g$ for 10 min. Approximately 30,000 EVs were added to J774 macrophages in 10% FSC high glucose DMEM and live imaging was performed the following day (~16 h post-co-incubation) with a Zeiss LSM inverted 880 confocal microscope with a 63x oil objective (1.4 NA). Signal was collected with internal detectors (GFP excitation = 488 nm, CypHer-5E excitation = 633 nm).

### Ex vivo AV blocking assay
BM from Flk1-GFP mice was harvested and centrifuged at 300 × $g$ for 10 min to pellet cells. The supernatant was collected and centrifuged at 3000 × $g$ for 20 min to pellet Flk1-GFP$^+$ EVs. The EV-enriched fraction was incubated with either 5 μg or 10 μg purified AV (BioLegend, #640902) for 1.5 h at 37 °C. BM from non-transgenic wild-type (wt) mice was harvested and centrifuged at 300 × $g$ for 5 min. Cells were stained with Cell Trace Violet (ThermoFisher Scientific, #C34557), washed three times and added to the EV-enriched fraction at a ratio of 1:4 per bone for 1 h at 37 °C. Cells were harvested, stained with TO-PRO-3 and CD45-PE and analysed by flow cytometry as described in the 'Methods' section 'Fluorescence activated cell sorting (FACS) and flow cytometry of murine samples'.

### Rapamycin treatment
For in vivo rapamycin treatment, Flk1-GFP mice were treated three times weekly with 4 mg/kg rapamycin (Merk Millipore, #553210) or PBS, administered via intraperitoneal injection for two weeks as previously described[10].

### Confocal microscopy of Flk1-GFP$^+$ EVs and cells
To assess the morphology and diameter of Flk1-GFP$^+$ EVs, EVs were isolated by differential centrifugation and FACS as described in the 'Methods' section 'Fluorescence activated cell sorting (FACS) and flow cytometry of murine samples' and added to an 8-well chamber slide (Ibidi, #80826). To visualize GFP fluorescence in immune cells, GFP$^+$ CD45.2$^+$ immune cells were isolated by FACS and added to an 8-well chamber slide (Ibidi, #80826). Isolated cells and EVs were imaged using a Zeiss LSM 880 inverted Confocal Microscope. Images were acquired using a Plan-Apochromat 63x 1.4 NA objective lens with a voxel size of 0.16 × 0.16 × 0.64 μm. Signal was collected with internal detectors (GFP excitation = 488 nm).

To assess mitochondria content in Flk1-GFP$^+$ EVs, EVs were isolated as described in the 'Methods' section 'Fluorescence Activated Cell Sorting (FACS) and flow cytometry of murine samples' and added to poly-D-lysine coated, glass bottom 8-well chamber slides (Ibidi, # 80827). Airyscan confocal microscopy was performed on a Zeiss LSM 880 with Airyscan module. Images were collected using a 1.4 NA 63x Plan-Apochromat lens with a voxel size of 0.04 × 0.04 × 0.17 μm. Light from 488 nm and 594 nm laser lines were directed via a 488/594 main beam splitter. Resultant fluorescence was detected via a plate with no additional filtering in place. All Airyscan images were processed using ZEN Black 3.1 using the auto-strength settings. Images are shown as selected 2D cross-sections from acquired z stacks.

### Calvarium intravital microscopy

Intravital microscopy of the calvarium BM was performed as previously described[42–44]. Briefly, mice were anesthetised with either isoflurane (4% isoflurane in 4 L/min O$_2$ for induction and 1–2% isoflurane in 1 L/min O$_2$ for maintenance) or by intraperitoneal/subcutaneous injection of 100 mg/kg Ketamine and 10 mg/kg Xylazine. The central portion of the scalp was removed and a custom headpiece was attached to the exposed calvarium using Ketac™ Cem Easymix™ dental cement (3M, #56900). The protective membrane covering the calvarium was removed and the headpiece was attached to a custom holder and secured for imaging. For live imaging experiments, protective eye gel (POLY VISC® Lubricating Eye Ointment, Alcon) was applied and body temperature was monitored by a digital rectal thermometer (PhysioSuite for Mice and Rats, Kent Scientific). Imaging was performed using an Olympus FVMPE-RS two-photon upright microscope equipped with a motorised stage and two tuneable infrared multiphoton lasers (Spectra physics IN-SIGHT X3-OL 680–1300 nm laser and Mai Tai HPDS-OL 690–1040 nm) with 25× magnification water-immersion lens (1.05 NA) and non-descanned detectors. Where specified, 100 μL of AV-BV605, AV-APC or Evans Blue was injected intravenously immediately prior to image acquisition. The following configurations were used for image acquisition. Collagen bone second harmonic generation (SHG) signal, Tomato and GFP signals were detected as follows: SHG excitation = 900 nm; emission = 410–455 nm. GFP excitation = 900 nm; emission = 495–540 nm. Tomato excitation = 1090 nm; emission = 575–645 nm. For in situ visualization of AV, BV605 was excited at 900 nm and detected at 575–645 nm (AV-BV605); APC was excited at 1025 nm and detected at 660–750 nm (AV-APC). To visualize the vasculature network, Evans Blue was excited at 860 nm and detected at 575–645 nm. Calvarium tile scans were acquired by a 7 × 4 tile scan, acquiring 5 μm z stacks. Tile time-lapse data was performed by acquiring 3 × 2 tile at 1.2 zoom, and 5 μm z stacks.

### Dual multiphoton confocal microscopy of undecalcified long bones

Long bones were analysed as previously published[42,43]. Briefly, mouse long bones (tibia and femurs) were harvested and fixed in Periodate-lysine-paraformaldehyde (PLP) for 24 h. Bones were washed in 0.1 M phosphate buffer, cryoprotected in a staggered sucrose gradient from 10% through to 30% (w/v) for 48–72 h and frozen in optimal cutting temperature (OCT) medium. Frozen samples were stored at −80 °C before sectioning with the NX70 Cryostat. For immunofluorescence analysis, 20 μm sections were obtained by the CryoJane tape transfer system (Leica microsystems). Slides were re-hydrated in PBS, permeabilised in 0.1% (v/v) Triton X-100 and blocked in 5% (v/v) goat serum for 30 min before incubation with primary antibody (Cleaved Caspase 3, Cell Signalling Technology, #9661) overnight at 4 °C. Slides were washed in PBS and incubated with secondary antibodies (goat anti-Rabbit AF633, ThermoFisher, #A-21070) in 0.1% (v/v) Triton X-100 for 1 h at RT. Stained slides were washed in PBS and mounted with Prolong Diamond antifade (Invitrogen, #P36965). Whole bone clearing

was performed as previously described[95]. Briefly, frozen bones were sectioned on either side to expose the BM cavity, washed overnight in PBS and incubated in RapiClear 1.52 (SunJin, #RC152002) for 3–7 days[95]. All bone images (sections and whole cleared bones) were obtained using a Zeiss LSM 880 upright confocal microscope equipped with Argon (458, 488 and 514 nm), a diode-pumped solid-state 561 nm laser and a Helium-Neon 633 nm, a tuneable infra-red multiphoton laser (Spectraphysics Mai Tai DeepSee 690–1020 nm). Signals were obtained with 4 non-descanned detectors (NDD) and an internal spectral detector array. External NDD detectors and the internal spectral array was separated using a 450 nm dichroic. SGH was excited 838 nm and detected on external NDD detector using a 460–485 nm filter. All other signals were collected with internal detectors as follows: GFP excitation = at 488 nm, Tomato excitation = 561 nm, goat anti-Rabbit AF633 excitation = 633 nm. Data was sampled using a W Plan-Apochromat 20x DIC water-immersion lens (1.0 N.A.).

### Zebrafish husbandry

Adult TU (Tübingen) wt and Tg(kdrl:mCherry)[96] zebrafish were housed at the LARTF, La Trobe University and maintained at 28 °C on a 12 h light/12 h dark cycle. All zebrafish work was conducted under AEC19002 as approved by La Trobe University Animal Ethics Committee. Embryos were collected in E3 embryo medium (5 mM NaCl, 0.17 mM KCl, 0.33 mM CaCl$_2$, 0.33 mM MgSO$_4$) and treated with 0.003% (w/v) 1-phenyl-2-thiourea (PTU; Sigma, #P7629) from 24 h post-fertilization (hpf) to inhibit pigmentation.

### Confocal microscopy of zebrafish-derived EVs

Embryos at 72 hpf were anesthetized with tricaine (Sigma, #E10521), and mounted in 0.8% (w/v) low melting point agarose in a glass bottom dish (ibidi, #81158) as previously described[97]. Time-lapse microscopy was performed using an LSM780 Confocal Microscope (Zeiss) with 25x objective (water, N.A. 0.8) with 4 μm z stacks and ~5 min intervals. Kdrl-mCherry$^+$ EVs were manually quantified using Fiji/Image J software. Where specified, autophagy was inhibited by treating de-chorionated larvae with 2.5 μM chloroquine for ~14 h[53].

### Flow cytometry analysis of zebrafish-derived EVs

Zebrafish embryos were prepared for flow cytometry as previously described[97,98]. In brief, 5–10 embryos were pooled together and chilled on ice. Embryos were rinsed in sterile calcium-free Ringer's solution with 2.5 mM EDTA for 15 min on a roller at 4 °C. Samples were gently resuspended to deyolk embryos and rinse solution was replaced with 0.25% (w/v) trypsin/EDTA for 15 min at 28 °C. Samples were resuspended to disassociate embryos and 1x PBS / 1% (w/v) BSA was added to inactivate trypsin. Single-cell solutions were filtered through a 40 μm cell strainer and centrifuged at 3000 × $g$ for 5 min to pellet cells and EVs. AV, MitoTracker Green and FLICA Caspase 3/7 staining was performed as above and acquired on a Beckman Coulter CytoFLEX.

### Image analysis

Image analysis was performed using Fiji[99,100] (version 2.9.0) and Imaris (Bitplane Scientific Software). Calvarium blood vessels were quantified by subtracting SHG signal from GFP on individual z stack images when necessary to remove signal crosstalk. GFP positive cells were selected using a manual image threshold. GFP$^+$ pixels were identified based on thresholding and a total pixel count was performed. Endothelial cells in cleared long bones were quantified by calculating the mean number of GFP$^+$ pixels relative to the total volume imaged. Mean values rather than absolute values were used to normalize bones of differing widths. For the quantification of blood vessel density in BM, FIJI was used to threshold Flk1-GFP signal of maximum intensity projected images and the area occupied by Flk1-GFP signal was then divided by the total area of BM. Cleaved caspase 3 / Flk1-GFP co-localisation was determined by binarizing corresponding image channels and measuring the area

fraction of caspase 3 within Flk1-GFP in 3D. A machine learning classifier was trained in ilastik[101] to detect caspase3 positive areas, whereas Flk1-GFP was binarized by applying a Gaussian blur followed by Otsu threshold using CLIJ2[102] in Fiji.

## Statistics and reproducibility

Raw data was visualized and processed using Microsoft Excel and GraphPad Prism (GraphPad Software Inc.). Outliers were identified by performing a ROUT Outlier test in GraphPad Prism. Where specified, statistical significance was determined by performing an unpaired two-tailed Student's $t$-test where significance was considered as $*p < 0.05$, $**p < 0.01$ and $***p < 0.001$ (ns = non-significant, $p \geq 0.05$). Independent repeats and number of animals used can be found in figure legends. No statistical method was used to predetermine sample size. For all in vivo experiments, mice were preselected based on age and then randomly assigned to treatment groups with gender distributed equally where possible. Vessel density and cleaved caspase 3 quantification was performed blinded. All attempts of replication which were not confounded by technical error were successful and as such, data represents a pool of independent repeats unless otherwise specified.

## Reporting summary

Further information on research design is available in the Nature Portfolio Reporting Summary linked to this article.

## Data availability

All data associated with this study are present in the main text or the Supplementary Information. Source data are provided with this paper.

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

## Acknowledgements

We thank Kelly Rogers and the WEHI Centre for Dynamic Imaging, as well as Chad Johnson and the La Trobe University BioImaging platform for their assistance with all microscopy experiments and analysis. We also thank ARAFlowCore within the Alfred Research Alliance at Monash University and the Cytometry & Advanced Fluorescence Imaging Nodes and the Materials Characterisation and Fabrication Platform at the University of Melbourne for assistance with imaging flow cytometry. We would like to acknowledge the contribution of the WEHI flow cytometry facility and thank the staff at the LARTF and the WEHI Bioservices Facility for maintaining zebrafish and mouse lines. This work has been supported by the National Health and Medical Research Council (2009287 to GAS, 1140187 to IP, 1173662 to IP, 2008652 to EDH), Australian Research Council (230101056 to IP, DP190102612 to WDF and EFL), CASS Foundation (9354 to GAS), Jack Brockhoff Foundation (4852 to GAS), L'Oreal UNESCO For Women in Science (to GAS), Victorian Cancer Agency (MCRF19045 to EFL), Sir Clive McPherson Family Fellowship (to VLB) and Rae Foundation grant (to VLB).

## Author contributions

GAS performed all experiments with help from J.P.S., A.L., J.S.R., A.L.S., J.E., J.L., P.R., N.D.G., R.K.H.Y., C.C., S.T., D.C.O. and E.D.H. D.N.J. and M.L. aided imaging flow cytometry. G.A.S., I.K.H.P. and E.D.H. designed all experiments. M.D.B. and T.K. designed and generated Cx3CR1-MerTK mice. M.C.K. provided zebrafish expertise and supervision. E.F.L. and W.D.F. provided autophagy expertise. V.L.B., M.M. and A.H.W. aided human work. K.L.R. provided microscopy support. G.A.S. wrote the manuscript with help from I.K.H.P. and E.D.H. and input from all other authors.

## Competing interests

The authors declare no competing interests.

## Additional information

[1]Walter and Eliza Hall Institute of Medical Research, Melbourne, VIC, Australia. [2]Department of Medical Biology, The University of Melbourne, Melbourne, VIC, Australia. [3]Department of Biochemistry and Chemistry, School of Agriculture, Biomedicine and Environment, La Trobe Institute for Molecular Science, La Trobe University, Melbourne, VIC, Australia. [4]Research Centre for Extracellular Vesicles, La Trobe University, Melbourne, VIC, Australia. [5]Clinical Haematology Department, Peter MacCallum Cancer Centre and Royal Melbourne Hospital, Melbourne, VIC, Australia. [6]Materials Characterisation and Fabrication Platform, Department of Chemical Engineering, University of Melbourne, Parkville, VIC, Australia. [7]ARAFlowCore, Alfred Research Alliance, Monash University, Melbourne, VIC, Australia. [8]Department of Clinical Immunology and Allergy, Royal Melbourne Hospital, Melbourne, VIC, Australia. [9]Department of Rural Clinical Sciences, La Trobe Rural Health School, Bendigo, VIC, Australia. [10]Florey Institute of Neuroscience and Mental Health, Melbourne, VIC, Australia. [11]Florey Department of Neuroscience and Mental Health, University of Melbourne, Melbourne, VIC, Australia. [12]Olivia Newton-John Cancer Research Institute, Heidelberg, VIC, Australia. [13]School of Cancer Medicine, La Trobe University, Bundoora, VIC, Australia. [14]These authors contributed equally: Edwin D. Hawkins, Ivan K. H. Poon. ✉e-mail: atkinsmith.g@wehi.edu.au; hawkins.e@wehi.edu.au; i.poon@latrobe.edu.au

