## [Peer Review File · Nature Communications]

In situ visualization of endothelial cell-derived extracellular vesicle formation in steady state and malignant conditionsEditorial Note:

This manuscript has been previously reviewed at another journal. This document only contains reviewer comments and rebuttal letters for versions considered at Nature Communications.

REVIEWERS' COMMENTS

Reviewer #1 (Remarks to the Author):

The authors have addressed the concerns satisfactorily.

Reviewer #2 (Remarks to the Author):

The authors have made significant efforts to address all of my concerns regarding direct observation of extracellular vesicle (EV) production in living tissue. However, I still have several comments:

1, In the introduction, some other papers that report on characterized extracellular vesicles carrying mitochondria, such as "mitopher" (PMID 37945830), are not included. The authors may want to provide a more comprehensive background on large vesicles that carry organelles.

2, Without understanding the biogenesis mechanism, it is not appropriate to define the observed EVs as "novel." Therefore, I strongly request the authors to remove the term "novel" from the manuscript. Instead, please just state the in-situ generation of large EVs from endothelial cells.

3, Since the extracellular vesicles identified in this study exhibit high levels of heterogeneity, the obtained time-lapse results are unable to differentiate the specific type of vesicle generated in the corresponding video. Therefore, the authors may want to adjust the textual description to account for this situation.

Reviewer #3 (Remarks to the Author):

The revised manuscript by Atkin-Smith et al along with authors' extensive rebuttals address some of the prior concerns. While the study is improved, somewhat extended and overall interesting, it does remain incomplete in some respects, as the authors mostly repel, rather than confront, the key questions surrounding the nature, biogenesis, mechanisms and biological significance of the large endothelial EVs they observe their experiments in vivo. In fact, as stated by the authors (in bold): "the main novelties of our study is the direct observation of EV production in living tissue under steady state and malignant conditions". This is unquestionably a technical accomplishment, albeit achieved by combining known technologies, but it gives the study a somewhat descriptive and observational flavour. Maybe deeper questions need to wait for another study, but they do remain unresolved.

What is the process leading to formation of large endothelial EVs? The authors provide little evidence that this is apoptosis or autophagy, they question the merit of dissecting EV heterogeneity on the grounds of MISEV guidelines (?) and they also dismiss suggestions that

a better molecular characterization of these EVs could reveal whether they may represent a new EV subtype. All this leaves the reader in a conceptual vacuum.

It is also unclear what is the role, function, or biological activity of large endothelial EVs described in the paper. There is an interesting correlation between the increase in circulating endothelial EVs and “vascular degradation” in cancer, but it is unclear what this ‘degradation’ means in biological terms. If this is neither autophagy, nor apoptosis, and there is an apparent loss of endothelial cells (?), perhaps its nature deserves some attention, and the related consequences for EV formation process (or processes) could be studied.

Likewise, the EV uptake (efferocytosis) by immune cells is described but the consequences of it remain unclear.

Because the authors never attempted to create a manageable (e.g. in vitro) model of large endothelial EV biogenesis, and they have not characterized them molecularly, many in-depth mechanistic questions are difficult to address, and remain unanswered.

It is understandable that discovery of a new type of EVs (if this is what it is) raise more questions than can be immediately addressed, but in a high profile submission, such as this, at least some of them could be attempted.

Thank you for handling our manuscript. We are thankful for the constructive feedback obtained from the referees. We have addressed all the points made by the referees herein.

Reviewer #1:

The authors have addressed the concerns satisfactorily.

Response: We thank the referee for their input into improving our manuscript.

Reviewer #2:

The authors have made significant efforts to address all of my concerns regarding direct observation of extracellular vesicle (EV) production in living tissue. However, I still have several comments:

1, In the introduction, some other papers that report on characterized extracellular vesicles carrying mitochondria, such as "mitopher" (PMID 37945830), are not included. The authors may want to provide a more comprehensive background on large vesicles that carry organelles.

Response: We have included recently published papers that are related to our work, such as the description of 'mitopher' (i.e. EVs harbouring organelles), in the introduction of our revised manuscript (line 75-77).

2, Without understanding the biogenesis mechanism, it is not appropriate to define the observed EVs as "novel." Therefore, I strongly request the authors to remove the term "novel" from the manuscript. Instead, please just state the in-situ generation of large EVs from endothelial cells.

Response: We have revised our manuscript accordingly to remove the term 'novel' and highlighted where appropriate the 'in situ generation of large endothelial cell-derived EVs'.

3, Since the extracellular vesicles identified in this study exhibit high levels of heterogeneity, the obtained time-lapse results are unable to differentiate the specific type of vesicle generated in the corresponding video. Therefore, the authors may want to adjust the textual description to account for this situation.

Response: We have highlighted in the revised that the EVs identified in this study are 'heterogenous mitochondria-rich EVs from endothelial cells...' (e.g. line 213-216).

Reviewer #3:

The revised manuscript by Atkin-Smith et al along with authors' extensive rebuttals address some of the prior concerns. While the study is improved, somewhat extended and overall interesting, it does remain incomplete in some respects, as the authors mostly repel, rather than confront, the key questions surrounding the nature, biogenesis, mechanisms and biological significance of the large endothelial EVs they

observe their experiments in vivo. In fact, as stated by the authors (in bold): “the main novelties of our study is the direct observation of EV production in living tissue under steady state and malignant conditions”. This is unquestionably a technical accomplishment, albeit achieved by combining known technologies, but it gives the study a somewhat descriptive and observational flavour. Maybe deeper questions need to wait for another study, but they do remain unresolved.

What is the process leading to formation of large endothelial EVs? The authors provide little evidence that this is apoptosis or autophagy, they question the merit of dissecting EV heterogeneity on the grounds of MISEV guidelines (?) and they also dismiss suggestions that a better molecular characterization of these EVs could reveal whether they may represent a new EV subtype. All this leaves the reader in a conceptual vacuum.

It is also unclear what is the role, function, or biological activity of large endothelial EVs described in the paper. There is an interesting correlation between the increase in circulating endothelial EVs and “vascular degradation” in cancer, but it is unclear what this ‘degradation’ means in biological terms. If this is neither autophagy, nor apoptosis, and there is an apparent loss of endothelial cells (?), perhaps its nature deserves some attention, and the related consequences for EV formation process (or processes) could be studied.

Likewise, the EV uptake (efferocytosis) by immune cells is described but the consequences of it remain unclear.

Because the authors never attempted to create a manageable (e.g. in vitro) model of large endothelial EV biogenesis, and they have not characterized them molecularly, many in-depth mechanistic questions are difficult to address, and remain unanswered.

It is understandable that discovery of a new type of EVs (if this is what it is) raise more questions than can be immediately addressed, but in a high profile submission, such as this, at least some of them could be attempted.

Response: We appreciate Reviewer #3’s view and we have addressed all these points and questions in our previous point-by-point response. Nevertheless, to further address certain unanswered questions beyond our current study, we have appropriately revised the discussion to highlight (line 422-426):

“Here, our study has focused on the visualisation of large EV production in situ and their subsequent interaction with immune cells under homeostatic conditions. However, fundamental questions such as how inflammation regulates the generation of large endothelial cell-derived EVs, the molecular mechanism(s) of their biogenesis and their functional impact on recipient cells remain to be fully defined and would be of significant interest in future studies.”